# Glutathione Deficiency and Alterations in the Sulfur Amino Acid Homeostasis during Early Postnatal Development as Potential Triggering Factors for Schizophrenia-Like Behavior in Adult Rats

**DOI:** 10.3390/molecules24234253

**Published:** 2019-11-22

**Authors:** Magdalena Górny, Agnieszka Wnuk, Adrianna Kamińska, Kinga Kamińska, Grażyna Chwatko, Anna Bilska-Wilkosz, Małgorzata Iciek, Małgorzata Kajta, Zofia Rogóż, Elżbieta Lorenc-Koci

**Affiliations:** 1The Chair of Medical Biochemistry, Jagiellonian University Medical College, 7 Kopernika Street, 31–034 Kraków, Poland; mbgorny@cyf-kr.edu.pl (M.G.); mbbilska@cyf-kr.edu.pl (A.B.-W.); miciek@cm-uj.krakow.pl (M.I.); 2Maj Institute of Pharmacology, Polish Academy of Sciences, 12 Smętna Street, 31–343 Kraków, Poland; wnuk@if-pan.krakow.pl (A.W.); k.kamin@if-pan.krakow.pl (K.K.); kajta@if-pan.krakow.pl (M.K.); rogoz@if-pan.krakow.pl (Z.R.); 3Department of Environmental Chemistry, University of Łódź, 163 Pomorska Street, 90-236 Łódź, Poland; adka367@interia.eu (A.K.); grazyna.chwatko@chemia.uni.lodz.pl (G.C.)

**Keywords:** social and cognitive deficits, GSH deficiency, global DNA methylation, neurodevelopmental model of schizophrenia, sulfur amino acids levels

## Abstract

Impaired glutathione (GSH) synthesis and dopaminergic transmission are important factors in the pathophysiology of schizophrenia. Our research aimed to assess the effects of l-buthionine-(*S*,*R*)-sulfoximine (BSO), a GSH synthesis inhibitor, and GBR 12909, a dopamine reuptake inhibitor, administered alone or in combination, to Sprague–Dawley rats during early postnatal development (p5–p16), on the levels of GSH, sulfur amino acids, global DNA methylation, and schizophrenia-like behavior. GSH, methionine (Met), homocysteine (Hcy), and cysteine (Cys) contents were determined in the liver, kidney, and in the prefrontal cortex (PFC) and hippocampus (HIP) of 16-day-old rats. DNA methylation in the PFC and HIP and schizophrenia-like behavior were assessed in adulthood (p90–p93). BSO caused the tissue-dependent decreases in GSH content and alterations in Met, Hcy, and Cys levels in the peripheral tissues and in the PFC and HIP. The changes in these parameters were accompanied by alterations in the global DNA methylation in the studied brain structures. Parallel to changes in the global DNA methylation, deficits in the social behaviors and cognitive functions were observed in adulthood. Only BSO + GBR 12909-treated rats exhibited behavioral alterations resembling positive symptoms in schizophrenia patients. Our results suggest the usefulness of this neurodevelopmental model for research on the pathomechanism of schizophrenia.

## 1. Introduction

Schizophrenia is a chronic, most devastating psychiatric illness affecting about 1% of the world population [1]. It develops progressively, often remaining undetected during childhood and adolescence, with the first episodes of psychosis that appear in early adulthood. The symptoms of the disorder are well characterized and divided into three main categories: positive symptoms (delusions, hallucinations, and thought disorder), negative symptoms (lack of motivation and deficits in social function), and cognitive deficits (impairment of attention, memory, and executive functions) [2]. It is increasingly recognized that schizophrenia is a neurodevelopmental disorder that involves abnormalities both in the structure and functioning of the brain. However, the mechanism underlying these pathological changes still remains poorly understood despite intensive studies. According to the prevailing hypothesis for the etiology of the disease, the structural and functional brain abnormalities could be initiated during embryonic or early postnatal development by numerous interactions between genetic and environmental factors [3], detrimental consequences of which appear in adulthood [4,5]. Moreover, since changes in the DNA sequence are only one aspect of the schizophrenia heritability, researches’ attention has recently been focused on epigenetic mechanism [6,7,8], i.e., modification of the regulation and expression of genes, without alterations in DNA sequence itself. These mechanisms include DNA methylation, histone modification, as well as regulations of RNA molecules [9]. Exploration of factors that initiate changes in DNA methylation [8,10,11] is particularly important because epigenetic regulation plays a crucial role in the brain development, synaptic plasticity, and formation of long-term memory [12,13,14,15].

Parallel to the above trends, a growing body of experimental data indicates that impaired redox-regulation and oxidative stress may play a significant role in the pathophysiology of schizophrenia [16,17,18,19,20,21]. In support of this view, it has been demonstrated that the level of glutathione (GSH), the main cellular non-protein antioxidant, and redox regulator [22,23] are decreased in the cerebrospinal fluid and medial prefrontal cortex (mPFC) of drug-naive schizophrenic patients [24,25] as well as in the post-mortem striatum and PFC of those treated earlier with antipsychotic drugs [26,27]. Interestingly, reduction of GSH concentration in schizophrenia seems to be linked to polymorphisms of genes encoding both catalytic and modifier subunits of γ-glutamate-cysteine ligase (GCL, EC 6.3.2.2.) [28,29,30], which is a key enzyme in GSH biosynthesis [31]. Furthermore, a significant negative correlation between the brain GSH levels [25] and the severity of negative symptoms was found in schizophrenic patients [32].

In rodents GSH levels can be depleted using the GCL, inhibitor l-buthionine-(*S*,*R*)-sulfoximine (BSO) [33], by using the GSH depleting agent diethyl maleate (DEM) [34], or via manipulation of genes encoding catalytic or modifier subunits of GCL [35,36,37]. Using these approaches some authors have reported on schizophrenia-like symptoms in animals showing numerous morphological and biochemical abnormalities reminiscent of those observed in patients with schizophrenia [16,17,18,21,38]. Consistently, referring GSH deficiency to schizophrenia-like symptoms, BSO given chronically into the cerebral ventricles of adult rats prior to dopamine (DA) administration, produced spatial memory deficits observed in the Morris water maze [39]. Acute, systemic administration of DEM, induced a severe reduction in the hippocampal GSH content and impaired short-term synaptic interactions (paired-pulses) and long-term potentiation without affecting baseline transmission [40]. Treatment of adult rats with DEM slowed down the acquisition of spatial learning in the Morris water maze [41]. Furthermore, chronic, administration of BSO in combination with an inhibitor of DA transporter (DAT), the compound GBR 12909, during the early postnatal development (p5–p16) to Osteogenic Disorder Shionogi (ODS) mutant rats, induced the long-term cognitive deficits, assessed in the novel object recognition test (NOR) in adulthood [42,43]. Correspondingly, treatment of ODS and Wistar rats during early postnatal life with BSO alone, evoked impairments of some cognitive functions when learning and discrimination tasks aimed at assessing spatial working memory performance, were examined in adulthood in the presence of controlled olfactory cues [44]. The abovementioned clinical and experimental data seem to confirm the important role of the GSH deficit in the pathogenesis of schizophrenia, but the molecular mechanisms underlying the pathological changes still remain unclear.

Synthesis of GSH is dependent on both GCL activity, the rate limiting enzyme involved in GSH biosynthesis, and on substrate availability including the amino acid cysteine (Cys) [45,46]. Cys is a semi-essential protein amino acid delivered in the diet and also synthesized endogenously via the transsulfuration pathway from homocysteine (Hcy), which is a key junction metabolite in methionine (Met) metabolism [31,47,48]. Moreover, Hcy apart from being metabolized via transsulfuration pathway can be methylated back to Met [49,50]. The above-described metabolic links between sulfur amino acids suggest that inhibition of GSH synthesis may lead to disturbances in the functioning of transsulfuration and re-methylation reactions, and consequently, to alterations in the concentrations of Met, Hcy, and Cys in different tissues. Interestingly, it was reported that treatment with Met caused a recrudescence of psychotic symptoms in schizophrenia patients [51], and schizophrenia-like symptoms in mice [10,11,52,53]. It is also worth remembering that S-adenosylmethionine (SAM) formed as the first metabolite during the conversion of Met into Hcy, [47], is the principal methyl donor that is crucial for methylation of key molecules involved in gene expression [6,54]. The most obvious example is DNA-methylation in cytosine-guanine (CpG) islands located in promoter and regulatory regions of numerous genes, which controls their transcription [6,8].

In the light of these data, it seems reasonable to suppose that inhibition of GSH synthesis in the rat body during early postnatal life may be a triggering factor for a cascade of events that starting from disturbances in the physiological homeostasis of sulfur amino acids and through further changes in the epigenetic status of some genes in the brain, can ultimately lead to the disclosure of schizophrenia-like symptoms in adult rats. To confirm this hypothesis, we performed a comprehensive analysis of the effects of chronic treatment with BSO and GBR 12909, on the content of GSH and sulfur amino acids (Met, Hcy, and Cys) and the global DNA methylation status, as well as on the expression of schizophrenia-like behaviors in Sprague–Dawley rats. The rationale for administration of the DAT inhibitor, compound GBR 12909, was to check whether the increase in the brain level of extracellular DA during the early postnatal development is sufficient to reveal positive symptoms of schizophrenia in adulthood, or whether simultaneous disturbances in dopaminergic transmission and GSH synthesis are required for their occurrence. Like in earlier studies [42,43,44], both BSO and GBR 12909 were given between postnatal day 5 and 16, alone or in combination. It is worth noting that at that time both the rate of rat brain growth and the physiological concentration of the brain GSH were reported to be the highest [55]. Concentrations of GSH, Met, Hcy, and Cys were measured in peripheral organs, which play a particular role in the metabolism of GSH and sulfur amino acids, i.e., the liver and kidney as well as in the brain, especially in its structures involved in the expression of schizophrenia-like symptoms, i.e., the PFC and hippocampus (HIP) in which relationships between the GSH synthesis and metabolism of sulfur amino acids are unclear and virtually unexplored. These parameters were determined in tissue homogenates from 16-day-old Sprague–Dawley males, 4 h after the last BSO, and/or GBR 12909 doses. The global DNA methylation in the PFC and HIP was measured twice, 4 h after the last doses of the model substances and in adulthood (at 93 days of age). Furthermore, in adult rats between days 90–92, schizophrenia-like behaviors, corresponding to the negative and positive symptoms as well as to cognitive deficits observed in schizophrenic patients, were assessed using appropriate tests. We hope that this set of experiments will shed a new light on the mechanisms underlying neurodevelopmental disturbances in schizophrenia.

## 2. Results

### 2.1. The Effects of Chronic Administration of BSO and GBR 12909 on the Levels of GSH and Sulfur Amino Acids in the Liver of 16-Day-Old Rats

Figure 1A–D illustrates the effects of chronic treatment with BSO and GBR 12909, alone or jointly, on the levels of GSH, Cys, Met, and Hcy determined in the liver of 16 days old male Sprague–Dawley pups, 4 h after the last doses of these compounds. See the Appendix A for the chromatograms of all the tissue homogenates of the rats.

A two-way analysis of variance (ANOVA) performed for GSH concentrations (Figure 1A) in the studied groups, revealed an overall treatment effect of BSO (F_(1,24)_ = 478.82; *p* < 0.00001), a lack of GBR 12909 treatment effect (F_(1,24)_ = 1.470; NS – non significant) and an interaction between these two model substances (F_(1,24)_ = 14.787; *p* < 0.001). Post hoc comparisons showed that BSO administered alone or especially in combination with GBR 12,909 decreased the GSH content when compared to the control group by 66.7% and 76.4%, respectively. In contrast to BSO, GBR 12909 alone increased the level of GSH by 18.7% of the control value (Figure 1A).

As to Cys levels in the studied groups (Figure 1B), a two-way ANOVA demonstrated no significant treatments effects of both BSO (F_(1,24)_ = 0.440; NS) and GBR 12,909 (F_(1,24)_ = 4.147; *p* = 0.052) and no interaction between these two compounds (F_(1,24)_ = 4.235; *p* = 0.051). Although there were no statistically significant differences in the Cys levels between the analyzed groups, some decreasing tendency in its content was observed in rats receiving GBR 12909 alone.

Regarding Met concentrations in the rat liver (Figure 1C), a two-way ANOVA revealed significant treatment effects of BSO (F_(1,24)_ = 13.808; *p* < 0.001) and GBR 12909 (F_(1,24)_ = 149.75; *p* < 0.00001) but no interaction between these compounds (F_(1,24)_ = 0.966; NS). In turn, the same analysis performed for the hepatic Hcy level (Figure 1D), showed only treatment effect of GBR 12909 (F_(1,24)_ = 91.495; *p* < 0.00001) but no effect of BSO (F_(1,24)_ = 1.892; NS) and no interaction between BSO + GBR 12909 (F_(1,24)_ = 0.025, NS). Post hoc comparisons showed that both BSO and GBR 12909 enhanced Met concentration in the rat liver (by 114% and 315% of the control, respectively), but the most pronounced increase in the Met level was observed when these compounds were administered jointly (by 381.5% of the control level; Figure 1C). In contrast to Met, the Hcy levels were significantly decreased in groups of rats receiving GBR 12909 alone or BSO + GBR 12909 compared to the control (by 65.1% and 75.3%, respectively; Figure 1D).

### 2.2. The Effects of Chronic Administration of BSO and GBR 12909 on the Levels of GSH and Sulfur Amino Acids in the Kidney of 16-Day-Old Rats

Similarly as in the liver, the changes in the levels of GSH, Cys, Met, and Hcy were examined in the kidney of 16-day-old pups treated chronically with BSO and GBR 12909, alone or jointly (Figure 2A–D).

A two-way ANOVA performed for GSH, Cys, and Met revealed a general treatment effect of BSO (for GSH F_(1,24)_ = 81.694, *p* < 0.00001; for Cys F_(1,28)_ = 285.51, *p* < 0.00001; for Met F_(1,28)_ = 12.576, *p* < 0.002) but a lack of treatment effect of GBR 12909 (for GSH F_(1,24)_ = 2.954, NS; for Cys F_(1,28)_ = 0.119, NS; for Met F_(1,28)_ = 1.371; NS) and no interactions between these model substances referring to these parameters (for GSH F_(1,24)_ = 0.308, NS; for Cys F _(1,28)_ = 0.441, NS; for Met F_(1,28)_ = 3.490; *p* = 0.07). As a two-way ANOVA for the Hcy level in the rat kidney was non-significant, the post-hoc analysis of this parameter was not performed (Figure 2D).

In general, the levels of GSH in the kidney of rats treated with BSO alone or in combination with GBR 12909 were distinctly reduced (by 49% and 30% of control value, respectively; Figure 2A). In parallel to the declines in GSH content, almost similar decreases in the Cys concentration were observed in these groups of rats (by 43% for BSO and 41% for BSO + GBR; Figure 2B) while Met levels increased significantly (by 32% and 27%, respectively, Figure 2C). Chronic administration of GBR 12,909 alone did not change significantly the levels of GSH and Cys, only Met content showed some increasing tendency in comparison to its value in the control group (Figure 2A–C).

### 2.3. The Effects of Chronic Administration of BSO and GBR 12909 on the Levels of GSH and Sulfur Amino Acids in the Brain Structures of 16-Day-Old Rats

A two-way ANOVA performed for GSH and Met concentrations in the PFC of male Sprague–Dawley pups treated chronically with BSO and GBR 12909, alone and jointly (Figure 3A,C) demonstrated a significant treatment effect of BSO (for GSH F_(1,28)_ = 28.72, *p* < 0.00001; for Met F_(1,28)_ = 45.331; *p* < 0.0001) but a lack of treatment effect of GBR 12909 (for GSH F_(1,28)_ = 0.10, NS; for Met F_(1,28)_ = 1.42; NS) and no interaction between BSO and GBR 12909 (for GSH F_(1,28)_ = 0.28, NS; for Met F_(1,28)_ = 2.31; NS) for these parameters. The same analysis carried out in four groups of 16-day-old rats for the Cys content (Figure 3B) revealed an overall treatment effects of BSO (F_(1,28)_ = 44.96, *p* < 0.00001) and GBR 12909 (F_(1,28)_ = 6.023, *p* < 0.05) as well as an interaction between these model substances (F_(1,28)_ = 12.568; *p* < 0.001). In turn, a two-way ANOVA carried out for Hcy concentration in the PFC showed a treatment effect of GBR 12909 (F_(1,28)_ = 7.536, *p* < 0.01), no treatment effect of BSO (F_(1,28)_ = 1.059, NS), and a lack of interaction between these two compounds (F_(1,28)_ = 3.636, *p* = 0.07).

Post-hoc analysis demonstrated that in the PFC, BSO alone or jointly with GBR 12909 caused small but statistically significant decreases in the GSH content (by ca. 7% of the control; Figure 3A). GBR 12909 alone did not affect GSH level in this structure. However, chronic administration of BSO, GBR 12909, or BSO + GBR 12909 markedly increased Cys concentrations by 47.5%, 28.5%, and 42.2% of the control value, respectively (Figure 3B). In turn, the Met level was reduced both in rats receiving BSO alone and jointly with GBR 12909 by 18.5% and 19.7% of the control value, respectively (Figure 3C) while Hcy concentration was increased only in the GBR 12909-treated group (Figure 3D).

In the HIP, a two-way ANOVA performed for GSH revealed a significant treatment effect of BSO (F_(1,28)_ = 12.12, *p* < 0.002) and GBR 12909 (F_(1,28)_ = 8.03; *p* < 0.01) as well as an interaction of BSO + GBR 12909 (F_(1,28)_ = 5.20; *p* < 0.05; Figure 4A). In the case of Cys, this analysis showed only treatment effect of BSO (F_(1,28)_ = 30.963; *p* < 0.000001) but a lack effect of GBR 12909 (F_(1,28)_ = 1.055; NS) and no interaction between model compounds (F_(1,28)_ = 3.565; *p* = 0.07; Figure 4B). A two-way ANOVA performed for the Met level in the HIP showed a lack of treatment effect of BSO (F_(1,28)_ = 0.194; NS) but a significant treatment effect of GBR 12909 (F_(1,28)_ = 7.888, *p* < 0.05), and an interaction between BSO and GBR 12909 (F_(1,28)_ = 27.622; *p* < 0.00001). This analysis performed for the hippocampal Hcy level revealed only a significant treatment effect of GBR 12909 (F_(1,28)_ = 30.936, *p* < 0.0001) but there was no treatment effect of BSO and no interaction between these compounds.

Post-hoc comparisons showed that in the HIP, BSO administered alone or jointly with GBR 12909 did not change the level of GSH, while GBR 12909 alone caused a small but statistically significant increase in its content (by 8.5% of the control value; Figure 4A). However, administration of BSO alone or in combination with GBR 12909 enhanced the concentration of Cys in this structure by 34.9% and 30.9% of the control value, respectively (Figure 4B). As to the Met level in the HIP, its concentrations in groups of rats receiving only BSO or GBR 12909 were significantly higher than in the control group by 37.5% and 62.7%, respectively. However, after the combined treatment, this effect was statistically non-significant when compared to control (Figure 4C). As to the Hcy content in the HIP, both GBR 12909 administered alone or jointly with BSO distinctly increased its level by 65.7% and 49.2% of the control value, respectively (Figure 4D).

### 2.4. The Effects of Chronic Administration of BSO and GBR 12909 on the Levels of the Global DNA Methylation in the PFC and HIP of 16-Day-Old Rats

The global DNA methylation in the PFC and HIP of 16-day-old rats was determined 4 h after administration of the last doses of BSO and GBR 12909, alone or jointly (Figure 5A,B).

In the PFC, a two-way ANOVA revealed a significant interaction of BSO + GBR 12909 (F_(1,19)_ = 15.258, *p* < 0.001) but no treatment effects of BSO (F_(1, 19)_ = 0.01, NS) or GBR 12909 (F_(1,19)_ = 0.23, NS) administered separately. In the HIP, the same analysis showed the overall treatment effects of BSO (F_(1,17)_ = 4.831, *p* < 0.05) and GBR 12909 (F_(1,17)_ = 5.435, *p* < 0.05) administered separately as well as an interaction of BSO + GBR 12909 (F_(1,17)_ = 7.456, *p* < 0.02). Post-hoc analysis showed that the global DNA methylation in the PFC of rats receiving BSO or GBR 12909 alone was significantly decreased while in group treated with BSO + GBR 12909 it was maintained at the control level (Figure 5A). In contrast to the PFC, in the HIP joint treatment with BSO + GBR 12909 increased the global DNA methylation when compared to the level of DNA methylation in the control group (Figure 5B).

### 2.5. The Effects of Chronic Administration of BSO and GBR 12909 during the Early Postnatal Life on the Expression of Schizophrenia-Like Behavior in Adulthood

Behavioral tests (social interaction test, SIT; novel object recognition test, NOR; and open field test, OFT) were performed in groups of adult rats (90–92 days old), which were treated with BSO and GBR 12909, alone and in combination between the postnatal days p5 and p16.

#### 2.5.1. Social Interaction Test (SIT)

Social behavior was assessed in 90 days old rats by means of two parameters, i.e., the total time spent in social interactions and the number of these interactions (Figure 6A,B).

A two-way ANOVA performed for the total time spent in social interactions revealed an overall treatment effect of BSO (F_(1,20)_ = 35.677, *p* < 0.001); a lack of treatment effect of GBR 12909 (F_(1,20)_ = 3.163, *p* = 0.09) and no interaction between these model compounds (F_(1,20)_ = 2.453, NS). The same analysis carried out for the number of interactions demonstrated only a significant treatment effect of BSO (F_(1,20)_ = 31.750, *p* < 0.001).

Post-hoc analysis showed that rats administered BSO alone or jointly with GBR 12909 spent much less time in social interactions (by 18% and 31.6%, respectively) when compared to value of this parameter in the control group (Figure 6A). Consequently, in both these groups the number of episodes (contacts) between two rats was also lower (by 29.5% and 39.3%) than between the control rats (Figure 6B).

#### 2.5.2. Novel Object Recognition Test (NOR)

In order to check whether BSO and GBR 12909 administered chronically, alone or jointly during the early postnatal life, induce the impairment of cognitive functions in adulthood, the NOR test was performed on the next day after SIT.

During the acquisition trial (session T1) rats from all examined groups spent equal time exploring two identical objects (A1 and A2; Figure 7A). In the retention trial (session T2) only control rats and those receiving GBR 12909 alone explored the novel object significantly longer than the familiar one (*p* < 0.001) while in groups treated with BSO alone or jointly with GBR 12909 these objects were explored with the similar intensity (Figure 7B). A two way ANOVA performed for the recognition index revealed a significant treatment effect of BSO (F_(1,36)_ = 35.677, *p* < 0.001) but a lack effect of GBR 12909 (F_(1,36)_ = 3.800, *p* = 0.059) and no interaction between these model compounds (F_(1,36)_ = 0.131, NS). Post hoc comparison showed that the values of the recognition indexes in groups of rats receiving BSO alone or jointly with GBR 12909 were significantly decreased when compared to the control group (by 24.6 and 30.1%, respectively; Figure 7C).

#### 2.5.3. Open Field Test (OFT)

On the 92nd day of postnatal life, the exploratory behavior in the open field (time of walking, ambulation, peeping, and rearing) was examined in all studied groups (Figure 8).

A two-way ANOVA for the time of walking revealed a significant treatment effect of BSO (F_(1,36)_ = 61.805, *p* < 0.0001), no treatment effect of GBR 12909 (F_(1,36)_ = 0.632, NS) and an interaction between BSO + GBR 12909 (F_(1,36)_ = 66.522, *p* < 0.0001). The same analysis carried out for the number of sector crossings (ambulation) also demonstrated a treatment effect of BSO (F_(1,36)_ = 30.048, *p* < 0.0001), a lack effect of GBR 12909 (F_(1,36)_ = 2.067, NS), and a significant interaction between these model compounds (F_(1,36)_ = 17.087, *p* < 0.0002) regarding this parameter. Only for the number of peeping and rearing episodes, a two way ANOVA showed significant treatment effects of both BSO (F_(1,36)_ = 19.024, *p* < 0.0001) and GBR 12909 (F_(1,36)_ = 7.183, *p* < 0.01) as well as a marked interaction (F_(1,36)_ = 19.952, *p* < 0.0001) between these compounds. Post hoc comparison of the studied groups showed that chronic combined administration of BSO and GBR 12909 during the early postnatal life (p5–p16) resulted in adulthood in the prolongation of the time of walking (Figure 8A) and in the increases in the numbers of sector crossings (Figure 8B) as well as the rearing and peeping episodes (Figure 8C). Interestingly, only in the group receiving GBR 12909 alone during the early postnatal development, the time of walking assessed in adulthood was distinctly decreased compared to the control and BSO-treated groups (Figure 8A).

### 2.6. The Effects of Chronic Administration of BSO and GBR 12909 during the Early Postnatal Life on the Levels of the Global DNA Methylation in the PFC and HIP in Adult Rats

After completing the above described behavioral tests, rats were killed and the global DNA methylation was determined in the PFC and HIP (Figure 9).

In the PFC, a two-way ANOVA revealed an overall treatment effect of BSO (F_(1,20)_ = 5.854, *p* < 0.05) and an interaction of BSO + GBR 12909 (F_(1,20)_ = 10.501, *p* < 0.01) but a lack treatment effect of GBR 12909 alone (F_(1,20)_ = 3.997, *p* = 0.059) while in the HIP, only a significant interaction of these model compounds was found (F_(1,20)_ = 11.423, *p* < 0.01) but there were no treatment effects of BSO (F_(1,20)_ = 3.591, *p* = 0.07) and GBR 12909 (F_(1,20)_ = 1.750, NS) alone. Post hoc comparisons showed that in the group of rats receiving BSO during early development, the global methylation of DNA assessed in adulthood (93^th^ day of age) was significantly enhanced in the PFC (by 45.5%) while in the HIP it was distinctly reduced (by 31.8%) when compared to the control group (Figure 9A,B).

## 3. Discussion

In order to facilitate interpretation of the results, the discussion was divided into sections and the potential mechanisms responsible for changes reported in this work described in relation schizophrenia patients. This division was to assist in reporting the effects of BSO and GBR 12909, administered alone or jointly, on GSH, Cys, Met, and Hcy contents in the examined peripheral tissues (liver and kidneys), and selected brain structures (PFC and HIP).

### 3.1. Impact of BSO and GBR 12909 on GSH Levels in the Studied Tissues

In our study, chronic administration of BSO and GBR 12909, alone or jointly, differently affected the levels of GSH measured in the liver, kidney, and the chosen brain structures in 16 days old rats. The most pronounced decreases in the GSH levels were observed in the liver, moderate in the kidney, and relatively small, but statistically significant, in the PFC. Interestingly, BSO treatment did not change the content of GSH in the HIP, keeping it at the level of the control group. The action of BSO both in the kidney and in the brain structures studied seems to indicate that some additional factors can modulate GSH synthesis in these tissues. These factors may include the tissue-specific utilization of Cys-containing dipeptides as direct substrates for GSH synthesis [46,56,57].

It is well known that BSO inhibits only the first step of GSH synthesis, that is the GCL- catalyzed reaction during which glutamate and Cys form the dipeptide γ-glutamylcysteine (γ-GluCys). The second step of GSH synthesis, i.e., the glutathione synthetase-(GS)-catalyzed reaction during which γ-GluCys is coupled with glycine thus forming GSH, is not inhibited by BSO. Despite the general rule that GSH is synthesized in all cell types from its precursor amino acids in a two-step reaction, the renal and brain cells can also use γ-GluCys as a direct substrate for GSH synthesis, and in this way, they can bypass the first reaction catalyzed by GCL [46,56,57,58]. In addition to the enzymatic activity of GCL, γ-GluCys formation in some tissues is closely linked to the activity of γ-glutamyl transpeptidase (γ-GT), the only extracellular-located enzyme that is capable of breaking down the extraordinary γ-peptide bond within the GSH molecule [59,60]. In the rat kidney, a very high activity of γ-GT, much higher than in other tissues, is mainly responsible for the maintenance of particularly high concentration of Cys [61,62], which in the extracellular space occurs in the disulfide form as cystine (Cys)_2_. During the extracellular degradation of GSH by renal γ-GT, the released γ-glutamyl moiety is transferred to the amino acid acceptor, i.e., (Cys)_2_, forming γ-glutamylcystine (γ-Glu(Cys)_2_) that after being taken up into cells is reduced to the dipeptide γ-GluCys, and then combined with glycine to form a GSH molecule in the reaction catalyzed by GS [56]. The above-described alternative pathway of GSH synthesis suggests that in the kidney even under conditions of GCL inhibition, the synthesis of this antioxidant may still be continued, but with a much lower efficiency due to a smaller pool of the extracellular GSH. This reasoning can explain why in the BSO-treated groups in our study, the decrease in GSH content in the rat kidney was significantly smaller than in the liver, which mainly uses precursor amino acids for GSH synthesis and is characterized by both much lower γ-GT activity and Cys concentration [59]. Consistently with the above result, it was demonstrated that acute administration of γ-GluCys in the presence of BSO enhanced the GSH level in the rat kidney [56]. Like in the kidney, the intracerebroventricular injection of γ-GluCys to the BSO-pretreated rats, also effectively increased GSH concentration in the substantia nigra and brainstem, but not in the cortex [63]. The latter study indicates that the use of γ-GluCys for GSH synthesis in the rat brain is dependent on the specificity of a given region of the brain.

Regarding the concentration of GSH in the brain, in the PFC of BSO-treated animals, a small but significant decrease in its content was observed, whereas in the HIP, GSH levels were unchanged. These results are in contradiction with earlier chronic BSO effects in 16-day-old mutant ODS rats, in which the same BSO dose as in our study, caused a nearly 50% drop in GSH content both in the PFC and HIP [64]. The reason for this discrepancy is unknown, but it is likely that Sprague–Dawley and ODS rats may differ in the basic permeability of the blood brain barrier (BBB) for BSO. On the other hand, it was demonstrated that in adult mice treated with a single dose of BSO, in response to the declines in GSH content in the liver and kidney, distinct increases in GSH levels were found either in the whole brain [65] or only in the cerebellum [66]. Considering our results obtained in 16-day-old Sprague–Dawley rats treated chronically with BSO, in the context of the above presented data, it is reasonable to suspect that the concentration of GSH in the PFC and HIP may have been modified in a specific manner depending on the structure, in response to a drastic drop in the hepatic and renal levels of GSH. As to the impact of GBR 12909 alone on the levels of GSH in the studied tissues, its chronic administration increased GSH concentration in the liver and HIP, while in the kidney and PFC, GSH was maintained at the control level. However, the effects of the combined treatment with BSO + GBR 12909 on GSH levels in these tissues were almost the same as that exerted by BSO alone.

### 3.2. Impact of BSO and GBR 12909 on Sulfur Amino Acid Levels in the Peripheral Tissues

The alterations in GSH content in the studied tissues were paralleled by distinct changes in the concentrations of sulfur amino acids, i.e., Cys, Met, and Hcy. Under physiological conditions, the liver takes up Cys and Met from the plasma as well as releases these amino acids to the circulation. In the human liver, almost 50% of Cys incorporated into the GSH molecule arises from the Met metabolite Hcy as a result of the transsulfuration reaction [49,67], while the remainder comes from the breakdown of proteins delivered in the diet.

#### 3.2.1. Modifications in Cys Concentrations

Concentrations of Cys determined in the liver of 16-day-old rats 4 h after the last chronic dose of BSO or combination of BSO + GBR 12909 were close to that in the control, although in such conditions, due to the insufficient use of this amino acid for GSH synthesis, an increase in the Cys content would be expected. Consistently with the above guess, Standeven and Wetterhahn [68] previously demonstrated that the hepatic Cys level was nearly doubled 20 min after acute BSO administration, but was not significantly different from control at later time points. However, in our study, regardless of the time of Cys determination, the lack of statistically significant changes in the Cys level in the liver of rats treated chronically with BSO alone paralleled by the increased Met content suggests shifting the balance between the transsulfuration and re-methylation of Hcy, in favor of re-methylation reaction. This shift in balance between these two metabolic pathways became even more evident in the group of rats treated chronically with GBR 12909 alone, in which Cys trended downward, while a significant increase in Met level was accompanied by a strong decline in Hcy content. Furthermore, in the GBR 12909-treated group an increase in GSH content was observed. In turn, combined administration of BSO + GBR 12909 caused both a much stronger Met level increase and Hcy decrease than did each of these model compounds administered alone. However, at such conditions, the level of Cys was slightly higher while GSH content was drastically reduced when compared to the group treated with GBR 12909 alone. The above-discussed data indicate that inhibition of GSH synthesis in the rat liver may lead to reduced activity of transsulfuration pathway in this tissue.

As to Cys concentration in the kidney of 16-day-old rats receiving a chronic dose of BSO alone or in combination with GBR 12909, Cys levels were significantly reduced when compared to the control. Under physiological conditions, the liver releases the substantial amount of GSH into the blood to supply other organs [69]. Indeed, the kidney removes about a half of the GSH exported by the liver, and due to having a high activity of γ-GT and dipeptidase, releases a comparable amount of Cys into the general circulation [61,69]. Since the decline in Cys content in the kidney of BSO-treated animals corresponded well with that of GSH, it is clear that degradation of GSH is the main source of Cys in the kidney. This effect also indicates that inhibition of GSH synthesis impairs liver-kidney inter-organ coordinated GSH metabolism.

#### 3.2.2. Modifications in Met and Hcy Concentrations

As to Met metabolism in the studied tissues, the liver is the main producer of GSH and an organ with the highest activity of Met metabolizing enzymes (see description to Figure 10). In the liver, at physiological conditions when Met is needed, re-methylation of Hcy to Met increases. Conversely, when Met is in excess, Hcy catabolism via the transsulfuration pathway is intensified [50,70]. The distribution of Hcy between the two competitive reactions provides a major regulatory locus for the metabolism of sulfur amino acids in the liver [49]. Therefore, one of the main roles of sulfur amino acid metabolism in the liver is to maintain the appropriate Met level [71,72].

In our study, high but varied increases in Met levels in the liver of rats treated with BSO or GBR 12909, alone or in combination, suggest that both inhibition of GSH synthesis and DA re-uptake lead to modification of Met metabolism. As to the impact of BSO alone on Met metabolism, it was demonstrated that in the liver of rats receiving a single dose of this compound, the MAT activity decreased by almost 60% [73]. Furthermore, a strong positive correlation (*r* = 0.936) between the GSH level and MAT activity presented in this study indicates that GSH plays an important role in the maintaining appropriate activity of the hepatic MAT. Finally, as a result of the reduced MAT enzymatic activity, it was found that the SAM level in the liver of BSO-treated rats decreased by 40% [73], suggesting abnormalities in methylation reactions in this organ. In line with these data, we can assume that in our study, an increased Met level in the liver of BSO-treated rats may result mainly from the reduced MAT activity. However, such explanation does not seem to be valid for GBR 12909-induced increase in the hepatic Met content, because at such treatment conditions GSH level increased. It is worth noting that in human liver dopamine (DA) is synthesized from tyramine by CYP2D6 isoenzyme (corresponding to CYP2D1 in the rat) [74]. An inhibition of DA re-uptake creates the need for Met, especially for its active metabolite SAM, which serves as a donor of methyl groups for catechol-*O*-methyltransferase (COMT) that metabolizes the extracellular DA to 3-methoxytyramine (3-MT). It should be added that in the mammalian body, the highest COMT activity was observed in the liver [75]. In our study, a significant rise in Met content in the rat livers receiving GBR 12909 alone, with a simultaneous decline in Hcy level, seems to indicate that the Hcy remethylation to Met has been intensified. In the light of these data, the highest increase in Met content observed in the liver of rats treated with BSO + GBR 12909 may result from both BSO-induced decline in MAT activity and increased Met synthesis by Hcy remethylation. In the latter group, along with the increase in the Met level, the strongest decline in Hcy content was found. Although Hcy is produced in all cells, its conversion to Met is differently modulated in tissues other than the liver. Consistently with the above, the increases in Met concentrations in the kidneys of rats of all studied groups, were accompanied by some slight downward trends in Hcy levels, indicating that Met synthesis via Hcy remethylation in this organ was less intense than in the liver.

A high plasma level of Met is found in several genetic disorders, mainly in MAT deficiency, which causes persistent hypermethioninemia [76]. Hypermethioninemia patients have exhibited neurological alterations often associated with abnormalities of brain myelination [77], that were also observed in schizophrenia patients [78,79]. Taking into account the above findings, we hypothesize that BSO-induced decrease in the hepatic activity of MAT during the early postnatal development and subsequent transient hypermethioninemia may be a potential factor leading to abnormal brain myelination and ultimately dysfunctional brain connectivity. Interestingly, it has been reported that cognitive deficits and clinical symptoms of schizophrenia can be induced via brain dysconnectivity [80,81].

### 3.3. Impact of BSO and GBR 12909 on Sulfur Amino Acid Levels in the Studied Brain Structures

In opposite to the liver and kidney, administration of BSO and GBR 12909, alone or jointly, differently affected concentrations of Cys, Met, and Hcy in the PFC and HIP.

#### 3.3.1. Modifications in Cys Concentrations

Both model compounds administered alone or jointly, increased the total Cys concentration in the PFC and HIP. The increase in this amino acid contents may be caused by increased import of the oxidized form of Cys, i.e., (Cys)_2_ from plasma into the brain [82] or by increased local Cys synthesis from Hcy via transsulfuration [83]. Although the transsulfuration pathway has been identified in the brain [83], an increase in Cys content via this route seems to be less likely, because activities of transsulfuration enzymes, i.e., CBS and especially CSE in this organ are relatively low, much lower than in the liver. Furthermore, in our study, in all examined groups of rats, despite increases in the Cys content both in the PFC and HIP, declines in Hcy levels were not observed. On the contrary, Hcy contents in these groups of rats trended upward or were statistically significantly enhanced when compared to the control. These data seem to confirm that the increased levels of Cys in the examined brain structures may result rather from its increased uptake from plasma than from the local synthesis.

Cys poorly penetrates the BBB, therefore, mainly (Cys)_2_, is transported into the brain. This transport is principally carried out by the Na^+^-independent (Cys)_2_/glutamate antiporter, also known as system x_c_^–^, that mediates the uptake of (Cys)_2_ in exchange for intracellular glutamate at a 1:1 ratio [84,85]. Inside the BBB, (Cys)_2_ is readily reduced and then the resulting Cys is transferred from the endothelial cells into the brain [86]. In the extracellular space, Cys is reoxidized to (Cys)_2_ and then transported, into glial cells, through the x_c_^–^ transport system [84,87]. In astrocytes, (Cys)_2_ is reduced to Cys, which is then either used for the GSH synthesis or transferred outside these cells [84]. Functional studies have indicated that system x_c_^–^ represents a highly inducible amino acid transport system [84]. Consistently with the above, in primary rat astrocyte culture, BSO significantly enhanced expression of the xCT subunit of the (Cys)_2_/glutamate antiporter, at protein and mRNA levels, as well as activity of this transport system [88]. So, we can suspect that in our study in rats treated chronically with BSO, alone or in combination with GBR 12909, the increased concentrations of the total Cys in the PFC and HIP, may be associated with an increased expression of the x_c_^–^ transport system.

Interestingly, an increase in the levels of xCT subunit of the (Cys)_2_/glutamate antiporter has been observed in the post-mortem study of the dorsolateral PFC obtained from patients with schizophrenia [89]. Furthermore, according to the study by Gysin et al. [30], schizophrenic patients having an impaired GSH synthesis, displayed a decreased total plasma Cys level but increased (Cys)_2_ content. In contrast to the latter study, Yang et al. [90] demonstrated that in schizophrenic patients the content of (Cys)_2_ in serum was decreased while its level in urine was elevated when compared to values of these parameters in controls. On the other hand, in a more recent study, Wang et al. [91] demonstrated that in schizophrenic patients, serum Cys levels were significantly higher than in control, and they were positively correlated with cognitive functions. There was no such correlation in healthy control patients. The existence of a cause and effect relationship between plasma Cys content and cognitive functions suggests that in our study the increase in the Cys level in the PFC and HIP, may be a compensatory mechanism to limit cognitive disturbances. The functional significance of Cys seems to be confirmed by clinical studies performed in schizophrenic patients, which showed the beneficial effects of *N*-acetyl-*L*-cysteine (NAC), an acetyl derivative of Cys, in alleviating negative symptoms [92,93,94].

#### 3.3.2. Modifications in Met and Hcy Concentrations

As to Met and Hcy concentrations in the PFC and HIP, their levels were modulated in a different manner by the used model compounds than in the liver and kidney.

In the PFC of rats receiving chronically either BSO alone or the combination of BSO + GBR 12909, significant decreases in Met and no significant changes in Hcy concentrations were observed compared to control group. However, in the PFC of rats treated chronically with GBR 12909 alone, some slight but statistically non-significant increase in Met content was accompanied by a significant increase in the Hcy level.

In the HIP of rats treated chronically only with BSO, the Met level was markedly enhanced but Hcy content was maintained at the control level. In turn, in rats receiving GBR 12909 alone, both Met and Hcy contents were significantly increased. However, in group of rats receiving BSO + GBR 12909, Met content did not differ significantly from that in control but Hcy was significantly enhanced. These data show that in the PFC and HIP under conditions of chronic treatment with BSO and GBR 12909, alone or jointly, metabolic relationship between Met and Hcy are much more complicated than in peripheral tissues.

It should be reminded that Hcy is produced by the hydrolysis of SAH in a reversible reaction catalyzed by SAHH (description to Figure 10). This reaction proceeds in the forward direction, as long as Hcy is efficiently removed via the remethylation or the transsulfuration pathways. However, under conditions when there is imbalance between these two metabolic pathways and Hcy content rises, its conversion into SAH is favored. SAH is not only the end product of many transmethylation reactions but also an inhibitor of different methyltransferases, including DNA methyltransferases, to which it binds with a higher affinity than SAM. Therefore, to assess the short- and long-term consequences of Met and Hcy content fluctuations in the studied brain structures, the analysis of the global DNA methylation was performed in the PFC and HIP, both in 16-day-old rats in the presence of model compounds, and in addition, in 93-day-old rats, after their withdrawal.

### 3.4. Impact of BSO and GBR 12909 on the Total DNA Methylation in the Studied Brain Structures

DNA methylation is one of the main epigenetic phenomena serving to establish and maintain tissue-specific pattern of gene expression. It plays a particularly important role in embryonic and postnatal development, therefore, disturbances in its course in these periods, may have functional consequences. Consistently, the locus-specific DNA hypomethylation is implicated in the etiology of various cancers and developmental syndromes. On the other hand, DNA hypermethylation that involves regions of CpG islands located at gene promoters may also be deleterious, as it results in transcriptional silencing the associated genes [95].

In our study, in the PFC of 16-day-old rats, both in the group treated with BSO or GBR 12909 alone, the global DNA methylation was significantly reduced, and only in the group receiving BSO + GBR 12909 its level was almost the same as in control. According to the study by Chamberlin et al. [96], hypomethylation can occur in the absence of changes in the Hcy concentration, under conditions in which the enzymatic activity of MAT is inhibited. On the other hand, Yi et al. [97] showed that chronic elevation in plasma SAH content, secondary to an increase in plasma total Hcy, strongly correlated with DNA hypomethylation in lymphocytes. In the light of these data, we can speculate that in our study chronic treatment with BSO alone reduces MAT activity not only in the liver [73] but also in the PFC and finally leads to hypomethylation. In turn, in rats treated chronically with GBR 12909, hypomethylation can be associated with enhanced SAH content consequent to the increased Hcy level.

In the HIP, the global DNA methylation was significantly increased in rats receiving jointly BSO + GBR 12909 while in groups treated with BSO or GBR 12909 alone, it was maintained at the control level. The obtained results indicate that model compounds exert a significant impact on the global DNA methylation, and their effects are specific for the studied brain structure. Although in the present study the functional consequences of these direct changes in the global DNA methylation were not further analyzed in 16-day-old rats, we can speculate that DNA hypo- or hypermethylation during early postnatal life may be somehow related to the deregulated expression of some genes, which in adulthood play an important role in revealing schizophrenia-like symptoms. Regarding the long-term effects of treatment with these model compounds, only the administration of BSO alone during early postnatal development led at adulthood to aberration in global DNA methylation, i.e., hypermethylation in the PFC and hypomethylation in the HIP.

### 3.5. Impact of BSO and GBR 12909 on the Schizophrenia-Like Behavior

The long-term alterations in the global DNA methylation pattern in the studied brain structures of BSO-treated rats occurred parallel to deficits in the social behavior and in cognitive functions, assessed in the SIT and NOR tests, respectively. The social and cognitive deficits were also observed in BSO + GBR 12909-treated group in which there were no significant changes in the global DNA methylation in the PFC and HIP. However, the lack of such changes in the global DNA methylation in this group of rats does not necessarily mean that the expression of at least, some specific genes has not been altered by hyper- or hypomethylation. In line with the latter assumption, our preliminary data showed that the brain-derived neurotrofic factor (BDNF) protein expression in the PFC and HIP of all groups of adult rats, which were treated with BSO or GBR 12909 during early postnatal life was markedly reduced in adulthood [98], suggesting hypermethylation of BDNF gene. The above-discussed behavioral results indicate that inhibition of GSH synthesis in Sprague–Dawley rats during early postnatal life is a key factor that initiates a cascade of events, involving changes in the metabolism of sulfur amino acids and DNA methylation, which in adulthood, lead to development of social deficits corresponding to negative symptoms and to cognitive impairment observed in schizophrenic patients. In contrast to BSO-treated groups, rats receiving only GBR 12909 did not show in adulthood either deficit in social behavior and cognitive functions or in the global DNA methylation. On the other hand, only rats treated in the early postnatal life with the BSO + GBR 12909 combination showed behavioral changes assessed in the OFT test in adulthood, corresponding to the positive symptoms seen in schizophrenic patients. This observation clearly indicates that only changes both in the content of GSH and dopaminergic transmission during the early postnatal life lead in adult rats to manifestation of behavioral changes reminiscent of positive symptoms of schizophrenia.

Since a growing body of experimental data shows the aberrant DNA methylation in schizophrenic patients [6,99], further studies are needed to establish which of the candidate genes suspected to be involved in the development of schizophrenia were aberrantly methylated in the neurodevelopmental model of this disease induced by administration of BSO alone or combination of BSO + GBR 12909. This experimental approach will allow us to determine the differences in methylation of genes responsible for negative symptoms and cognitive deficits from methylation of genes responsible for positive symptoms.

In closing, our results suggest that transient changes in the content of GSH and sulfur amino acid Met during the early postnatal life lead to alterations in epigenetic status in the PFC and HIP in adult rats, and to the occurrence of schizophrenic-type social and cognitive deficits. This is summarized in Figure 11.

## 4. Materials and Methods

The experiments were carried out in compliance with the Act on Experiments on Animals of January 21, 2005 amended on January 15, 2015 (published in Journal of Laws no 23/2015 item 266, Poland), and according to the Directive of the European Parliament and of the Council of Europe 2010/63/EU of 22 September 2010 on the protection of animals used for scientific purposes. They received also an approval of the Local Ethics Committee at the Institute of Pharmacology, Polish Academy of Sciences (permission no 3/2018 of 11 January 2018).All efforts were made to minimize the number and suffering of animals used.

### 4.1. Animals and Treatment

Pregnant Sprague–Dawley females at embryonic day 16 delivered by Charles River Company (Sulzfeld, Germany) were kept in an individual cages under standard laboratory conditions; at room temperature (22 °C) under an artificial light/dark cycle (12/12; lights on from 7 am, lights off from 7 pm), with free access to standard laboratory chow and tape water. On the day of birth, the sex of pups was determined, and only males were left with their mothers to be used in further experimental procedure. Between the postnatal days p5 and p16, male Sprague–Dawley pups were treated with BSO (3.8 mmol/kg s.c., daily), the selective inhibitor of GCL, and the DAT inhibitor GBR 12909 (5 mg/kg s.c., every second day), alone or in combination. Control pups instead of the BSO or GBR 12909 were given vehicle. Rats were weighed daily and the injected quantity was adjusted accordingly to the actual body weight. On postnatal day p23 rats were weaned and housed in groups of four to five until p92.

Experiments were performed on several groups of rats. Determination of the levels of GSH and sulfur amino acids (Cys, Met, and Hcy), in the peripheral tissues (liver and kidney), and in the brain structures (PFC and HIP) was performed in 16-day-old male pups. Behavioral tests (SIT, NOR, and OFT) evaluating expression of schizophrenia-like symptoms were carried out in adulthood (at p90–92 days of age). The levels of DNA methylation were determined in the PFC and HIP in both 16-day-old rats, which were killed 4 h after the last doses of the model compounds and in adult 93-day-old rats killed on the next day after the last behavioral test.

### 4.2. Chemicals and Reagents

GSH Assay Kit (Cayman Chemical Company, Ann Arbor, MI, USA, Item No. 703002) was purchased from Biokom (Warsaw, Poland), 1-[2-[Bis(4-fluorophenyl)methoxy]ethyl]-4-(3-phenylpropyl)piperazine dihydrochloride (GBR 12909) from Abcam Biochemicals (Cambridge, UK) while the GenElute™ Mammalian Genomic DNA Miniprep Kit and Imprint^®^Methylated DNA Quantification Kit from Sigma Aldrich Chemical Company (St. Louis, MO, USA).

l-buthionine-(*S*,*R*)-sulfoximine (BSO), l-cysteine hydrochloride (l-Cys*HCl), glutathione (GSH), methionine (Met), homocysteine (Hcy), *N*-acetyl-cysteine (NAC), *O*-phthaldialdehyde (OPA), tris-(2-carboxyethyl)phosphine (TCEP), trichloroacetic acid (TCA), 2,7-dichlorofluorescein (DCF), 2,7-dichlorofluorescein diacetate (DCFH-DA), *p*-phenylenediamine (pfda), iron(III) chloride (FeCl_3_), iron(III) chloride hexahydrate (FeCl_3_× 6H_2_O), iron(II) sulfate heptahydrate (FeSO_4_ × 7H_2_O), dithiothreitol (DTT), sodium hydrosulfide (NaHS), sodium sulfide (Na_2_S), sodium hydroxide (NaOH), sodium acetate, EDTA, potassium thiocyanate (KSCN), α-ketobutyric acid, 3-methyl-2-benzothiazolinone hydrazine hydrochloride monohydrate (MBTH), pyridoxal 5′-phosphate monohydrate (PLP), L-homoserine, 2,4,6-tris(2-pyridyl)-s-triazine (TPTZ), thionine, zinc acetate, albumin bovine, and gelatin were received from Sigma-Aldrich Chemical Company (St. Louis, MO, USA).

2-Chloro-1-methylquinolinium tetrafluoroborate (CMQT) was prepared in the Department of Environmental Chemistry, University of Łódź (Łódź, Poland) according to the procedure described by Bald and Głowacki [100].

Sodium hydroxide (NaOH), HPLC-grade acetonitrile (ACN), sodium hydrogen phosphate heptahydrate (Na_2_HPO_4_ × 7H_2_O), sodium dihydrogen phosphate dihydrate (NaH_2_PO_4_ × 2H_2_O) were from J.T. Baker (Deventer, The Netherlands), while perchloric acid (PCA), ninhydrin sodium, and potassium cyanide (KCN) were obtained from Merck (Darmstadt, Germany). Acetic acid (CH_3_COOH), barium chloride (BaCl_2_), hydrochloric acid (HCl), ammonia (NH_3_), formaldehyde, potassium dihydrogen phosphate (KH_2_PO_4_), sodium sulfate (Na_2_SO_4_), and Folin-Ciocalteu phenol reagent were purchased from Polish Chemical Reagent Company (P.O.Ch, Gliwice, Poland).

### 4.3. Biochemical Methods

#### 4.3.1. Preparation of Tissue Homogenates

The frozen tissues were weighted and immediately homogenized using an IKA-ULTRA-TURRAX T10 homogenizer (IKA Poland Sp. Z o.o company, Warsaw, Poland) at an operating speed of 6000 rpm. Tissue homogenates were prepared in two ways. The samples of brain structures (prefrontal cortex and hippocampus), liver, and kidney for HPLC analysis were homogenized in 0.2 M phosphate buffer, pH 8.2 at the ratio 1:10 (*w*/*v*, g/mL) for 0.5 min. The livers and kidneys for other analyzes were homogenized in 0.1 M phosphate buffer, pH 7.4 at the ratio 1:4 (*w*/*v*, g/mL) for 1 min. Homogenates were then used for biochemical assays.

#### 4.3.2. Determination of the Total GSH Content in Liver and Kidney Homogenates

Determination of the total GSH in the liver and kidney homogenates was carried out using the Glutathione Assay Kit (Cayman Chemical Company, Ann Arbor, MI, USA), which utilizes carefully optimized enzymatic recycling method for the quantification of GSH.

The sulfhydryl group of GSH reacts with Ellman’s reagent (5,5′-dithiobis-2-nitrobenzoic acid-DTNB) and produces a yellow colored 5-thio-2-nitrobenzoic acid (TNB). The second product of the reaction, mixed disulfide GSTNB, is reduced by GSSG reductase to recycle the GSH and produce more TNB. The rate of TNB production is directly proportional to the concentration of GSH in the sample. The absorbance of TBN is measured at 410 nm.

Briefly, 150 µL of freshly prepared assay mixture (NADPH, GSSG reductase, MES buffer consisting of 0.4 M 2-(*N*-morpholino)ethanesulfonic acid, 0.1 M phosphate, and 2 mM EDTA, pH 6.0, and DTNB) were added to 50 µL of homogenate after deproteinization. Incubation was carried out in the dark and the absorbance was read after 6 min. The GSH concentration was calculated from a standard curve prepared with 25 µM of GSSG and was expressed in nmoles of GSH per g tissue.

#### 4.3.3. Determination of the Cys Level in Liver and Kidney Homogenates

The levels of Cys were assayed by colorimetric reaction with ninhydrin solution [101]. To 0.95 mL of homogenate, 0.05 mL of 50% TCA was added, and centrifuged at 10,000× *g* for 10 min at 4 °C. Subsequently, to 0.125 mL of supernatant, the following reagents were added: 0.125 mL of 2.5% TCA, 0.125 mL of 99.5% acetic acid, and 0.125 mL of ninhydrin reagent (250 mg of ninhydrin, 6 mL of acetic acid, and 4 mL of hydrochloric acid). The reaction mixture was incubated in a boiling-water bath for 10 min, cooled, and then 0.5 mL of ethanol was added. Absorbance was measured at 560 nm. The Cys concentration was calculated from a standard curve prepared for 1 mM l-Cys*HCl and was expressed in nmoles per g tissue.

#### 4.3.4. Determination of Sulfur Amino Acids in the Rat Tissues Using the HPLC Method

##### Determination of Total GSH and Cys Levels in the Brain Structures

Assay procedure for tissues:

Total GSH and Cys levels in the brain samples (PFC and HIP) were analyzed by high-performance liquid chromatography (HPLC) with ultraviolet (UV) detection according to the method described by Bald et al. [102] with a modification developed by Kamińska et al. [103]. Simultaneous determination of thiols was carried out using HPLC after precolumn derivatization with CMQT [100].

Briefly, 100 µL of the appropriate brain structure tissue homogenate (PFC and HIP) was incubated with 7.5 µL of 0.25 M Tris-(2-carboxyethyl)phosphine (TCEP) solution at room temperature for 15 min in order to reduce disulfide bonds. Next, 10 µL of 0.1 M CMQT was added, vortexed and kept at room temperature for 5 min. The mixture was acidified with 15 µL of 3 M perchloric acid (PCA). The precipitated proteins were removed by centrifugation at 12,000 rpm for 10 min at 10 °C. The supernatant samples (10 µL) were injected (using an autosampler) into the HPLC system (1220 Infinity LC system from Agilent, Waldbronn, Germany) equipped with Zorbax SB-C18 column (Agilent Technologies), a diode-array detector and controlled by OpenLAB CDS ChemStation Edition software (Rev. C.01.05, Agilent Technologies, Waldbronn, Germany). Each sample was analyzed in duplicate. The mobile phase consisted of 0.1 M trichloroacetic acid (TCA) adjusted by 1M NaOH to pH = 1.6 (A) and acetonitrile (B). The flow-rate was 1 mL/min, temperature 25 °C and detector wavelength 355 nm. For separation of the 2-*S*-quinolinium derivatives of thiols from each other and from reagent excess the following gradient elution was used: 0–3.5 min, 11%–25% (B); 3.5–5.5 min, 25%–40% (B); and 5.5–9 min 40%–11% (B).

Identification of GSH and Cys peaks was based on the comparison of retention times and diode-array spectra with the corresponding set of data obtained for standard compounds. GSH and Cys concentrations were read from a calibration curve prepared from GSH (0–300 nmol/mL) and Cys (0–30 nmol/mL), and were expressed in nmoles GSH or Cys per g tissue, respectively.

##### Determination of Met and Hcy Levels in the Liver, Kidney, and Brain Structures

Assay procedure for tissues:

Met and Hcy levels were determined by HPLC method with fluorescence (FL) detection described by Borowczyk et al. [104] with some modifications. This HPLC-FL method is based on an on-column derivatization with *O*-phthaldialdehyde (OPA) and is used for the determination of thiols in biological samples.

Briefly, 100 µL of homogenate was treated with 14 µL of 0.25 M Tris-(2-carboxyethyl)phosphine (TCEP) solution for 10 min at room temperature in order to reduce disulfide bonds. Then, 30 µL of 0.5 M *N*-acetyl-cysteine (NAC) solution and 10 µL of 3 M perchloric acid (PCA) were added and the mixture was vortexed and centrifuged at 12,000 rpm at 10 °C for 10 min. The supernatant was transferred into HPLC system. HPLC analysis was performed on the HPLC system (1100 series from Hewlett-Packard, Waldbronn, Germany) equipped with FL detector (1260 Series from Agilent Technologies, Waldbronn, Germany) and controlled by HP ChemStation software (Rev. A.10.02, Hewlett-Packrd, Waldbronn, Germany). The samples of supernatant (5 µL) were injected into the column PRP-1 Hamilton (150 nm × 4.6 nm × 5 µm) and each sample was analyzed in duplicate. The mobile phase consisted of 0.01 M OPA in 0.1 M NaOH and acetonitrile in the gradient mode described earlier [104] and was applied at the flow rate 1 mL/min. Amounts of Met and Hcy were quantified by using an FL detector at two different excitation and emission wavelengths (0–8 min excitation at 348 nm and emission at 438 nm and 8–14 min excitation at 370 nm and emission at 480 nm). Identification of peaks was based on comparison of retention time with data obtained for standard compounds. The levels of Met and Hcy were evaluated using the calibration curves prepared for Met (0–20 nmol/mL) or Hcy (0–7 nmol/mL) and finally were expressed in nmoles Met and Hcy per g tissue, respectively.

#### 4.3.5. Measurement of the Global DNA Methylation

High-quality and ultra-pure genomic DNA (without any RNA contamination) was extracted from prefrontal cortex and hippocampus using the GenElute™ Mammalian Genomic DNA Miniprep Kit according to the manufacturer’s instructions. Global changes in DNA methylation were measured in the tested tissue samples using a specific ELISA-based Kit (Imprint^®^Methylated DNA Quantification). The quantity and quality of DNA was spectrophotometrically measured at 260 nm and 260/280 nm (ND/1000 UV/VIS; Thermo Fisher NanoDrop, Waltham, MA, USA) [105,106]. This kit contained all the reagents required to determine the relative levels of methylated DNA, including control methylated DNA. Methylated DNA was detected using capture and detection antibodies and quantified colorimetrically using an Infinite M200pro microplate reader (Tecan, Austria). The amount of methylated DNA present in the sample was proportional to the absorbance measured.

### 4.4. Behavioral Methods

#### 4.4.1. Social Interaction Test

The social interaction test (SIT) was performed using a black PCV box (67 cm × 57 cm × 30 cm, length × width × height). The arena was dimly illuminated with an indirect light of 18 Lux [107]. Each social interaction experiment involving two rats was carried out during the light phase of the light/dark cycle. The rats were selected from separate housing cages to make a pair for the study. The paired rats were matched for body weight within 15 g. Rats of each pair were placed diagonally in opposite corners of the box facing away from each other. The behavior of the animals was measured over a 10-min period. The test box was wiped clean between each trial. Social interaction between two rats was expressed as the total time spent in social behavior, such as sniffing, genital investigation, chasing, and fighting with each other. The number of episodes was counted as a separate paradigm. The SIT test was performed on day p90. Each group was composed of 12 rats (six pairs).

#### 4.4.2. Novel Object Recognition Test

The novel object recognition (NOR) test was performed using a black PCV box (67 cm × 57 cm × 30 cm, length × width × height). The arena was dimly illuminated with an indirect light of 18 Lux. On the first day of the experiment (adaptation), a rat was placed in the box for 10 min. On the next day, the animal was placed in the box for 5 min (T1) with two identical objects (white tin 5 cm wide and 14 cm high or green pyramid 5 cm wide and 14 cm high). The exploration time of objects was measured for each of the two objects separately. Then, one hour after T1, each rat was again placed in the box for 5 min (T2), with two different objects: one of the previous session (old) and the other new (white box and green pyramid). The exploration time of objects was measured for each of the two objects separately (sniffing, touching or climbing). The NOR test was performed on day p91. Each group was composed of 10 rats.

#### 4.4.3. Open Field Test

Exploratory activity was assessed in the elevated OFT. A black circular platform without walls having 1 m in diameter was divided into six symmetrical sectors and was elevated 50 cm above the floor. The laboratory room was dark and only the centre of the open field was illuminated with a 75 W bulb placed 75 cm above the platform. At the beginning of the test, the animal was placed gently in the centre of the platform and was allowed to explore. The exploratory activity, ambulation, peeping, and rearing in the open field, i.e., respectively, the time of walking, the number of sector crossings, and the number of episodes of peeping under the edge of arena and rearing were assessed for 5 min. The OFT test was performed on day p92. Each group consisted of 10 rats.

### 4.5. Statistics

The statistical analysis of the obtained biochemical and behavioral data was performed using a two-way ANOVA followed (if significant) by the Newman–Keuls test for post-hoc comparisons.

## Figures and Tables

**Figure 1 molecules-24-04253-f001:**
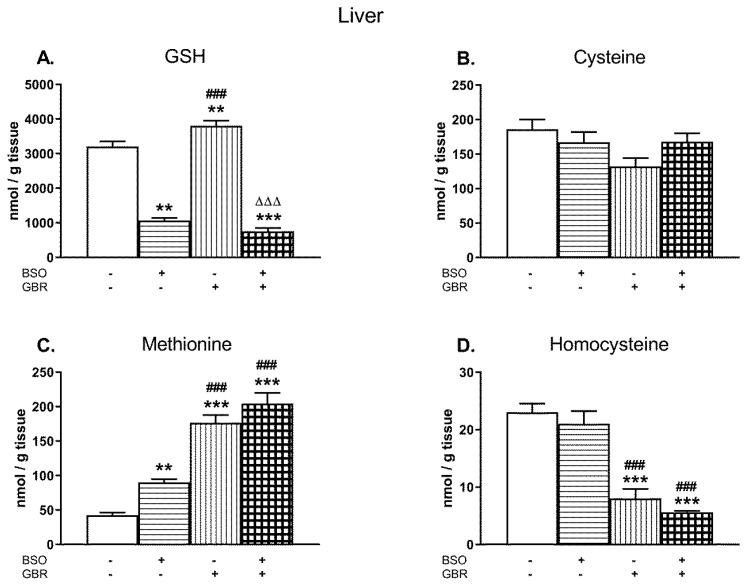
The effects of chronic administration of BSO (3.8 mmol/kg s. c., once daily) and the DAT inhibitor GBR 12,909 (5 mg/kg s. c., every second day), alone or in combination, on the concentrations of GSH (**A**), Cys (**B**), Met (**C**), and Hcy (**D**) in the liver of 16-day-old rats. Data expressed in nmole/g of tissue are presented as the mean ± SEM, *n* = 7 for each group. Statistical analysis was performed using a two-way ANOVA; symbols indicate significance of differences according to the Newman–Keuls post-hoc test, ***p* < 0.01, ****p* < 0.001 vs. control; ^###^*p* < 0.001 vs. BSO- and ^∆∆∆^*p* < 0.001 vs. GBR 12909-treated groups.

**Figure 2 molecules-24-04253-f002:**
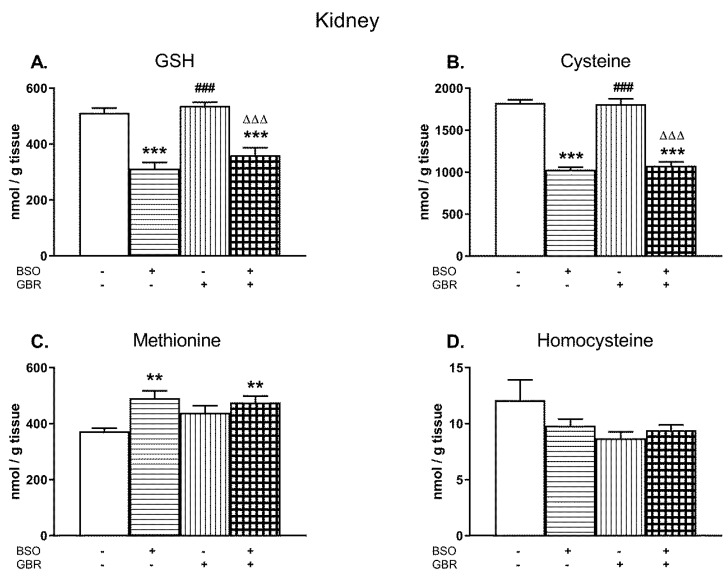
The effects of chronic administration of BSO and GBR 12909, alone or in combination, on the levels of GSH (**A**), Cys (**B**), Met (**C**), and Hcy (**D**) in the kidney of 16-day-old rats. Data expressed in nmole/g of tissue are presented as the mean ± SEM, *n* = 7–8 for each group. Statistical analysis was performed using a two-way ANOVA; symbols indicate significance of differences according to the Newman–Keuls post-hoc test, ***p* < 0.01, ****p* < 0.001 vs. control; ^###^*p* < 0.001 vs. BSO-; and ^∆∆∆^*p* < 0.001 vs. GBR 12909-treated groups.

**Figure 3 molecules-24-04253-f003:**
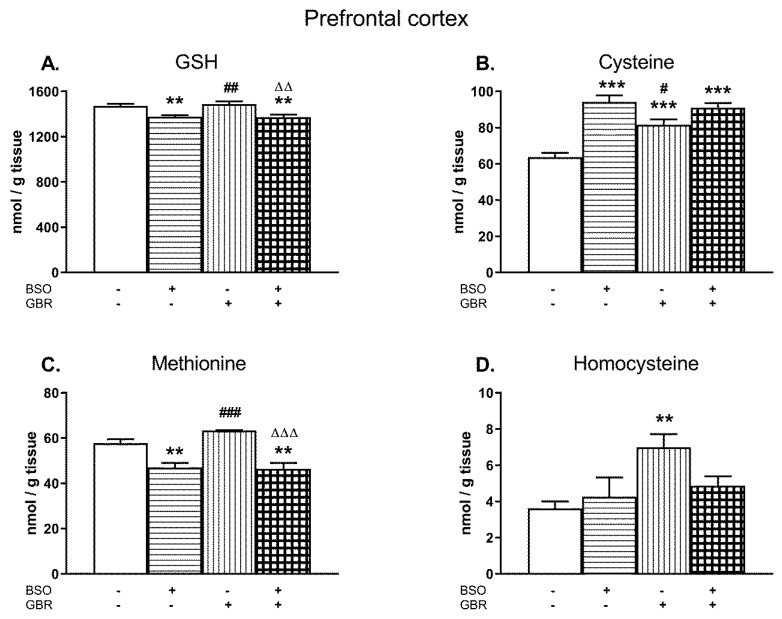
The effects of chronic administration of BSO and GBR 12909, alone and in combination, on the levels of GSH (**A**), Cys (**B**), Met (**C**), and Hcy (**D**) in the prefrontal cortex (PFC) of 16-day-old rats. Data expressed in nmole/g of tissue are presented as the mean ± SEM, *n* = 8 for each group. Statistical analysis was performed using a two-way ANOVA; symbols indicate significance of differences according to the Newman–Keuls post-hoc test, ***p* < 0.01, ****p* < 0.001, vs. control; ^#^*p* < 0.05, ^##^*p* < 0.01, ^###^*p* < 0.001 vs. BSO- and ^∆∆^*p* < 0.01, ^∆∆∆^*p* < 0.001 vs. GBR 12909-treated groups.

**Figure 4 molecules-24-04253-f004:**
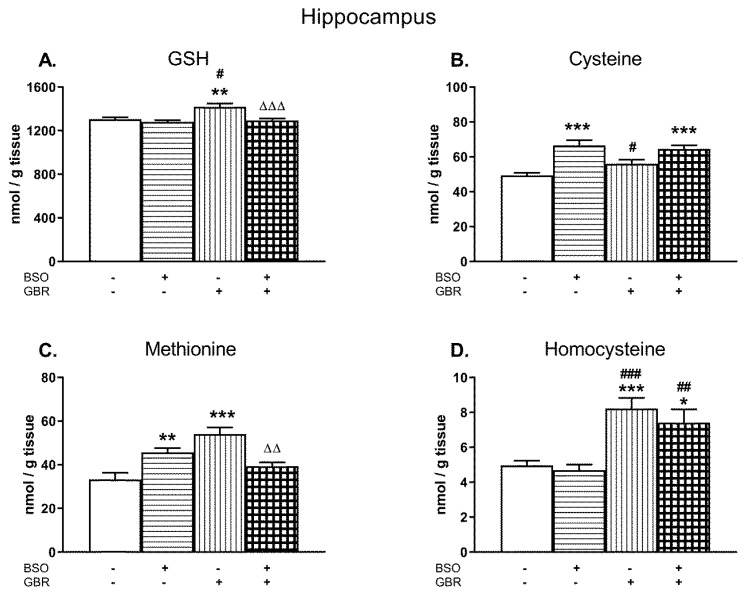
The effects of chronic administration of BSO and GBR 12909, alone or in combination, on the levels of GSH (**A**), Cys (**B**), Met (**C**), and Hcy (**D**) in the HIP of 16-day-old rats. Data expressed in nmole/g of tissue are presented as the mean ± SEM, *n* = 8 for each group. Statistical analysis was performed using a two-way ANOVA; symbols indicate significance of differences according to the Newman–Keuls post-hoc test, **p* < 0.05, ***p* < 0.01, ****p* < 0.001 vs. control; ^#^*p* < 0.05; *^##^p* < 0.01, ^###^*p* < 0.001 vs. BSO-; and ^∆∆^*p* < 0.01, ^∆∆∆^*p* < 0.001 vs. GBR 12909-treated groups.

**Figure 5 molecules-24-04253-f005:**
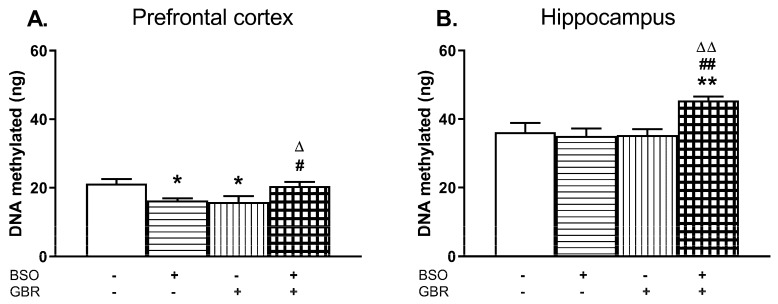
The effects of chronic administration of BSO and GBR 12909, alone or in combination, on the global DNA methylation in the PFC (**A**) and HIP (**B**) of 16-day-old rats. Statistical analysis was performed using a two-way ANOVA; symbols indicate significance of differences according to the Newman–Keuls post-hoc test, **p* < 0.05, ***p* < 0.01 vs. control; ^#^*p* < 0.05, ^##^*p* < 0.01 vs. BSO-; and ^∆^*p* < 0.05, ^∆∆^*p* < 0.01 vs. GBR 12909-treated groups.

**Figure 6 molecules-24-04253-f006:**
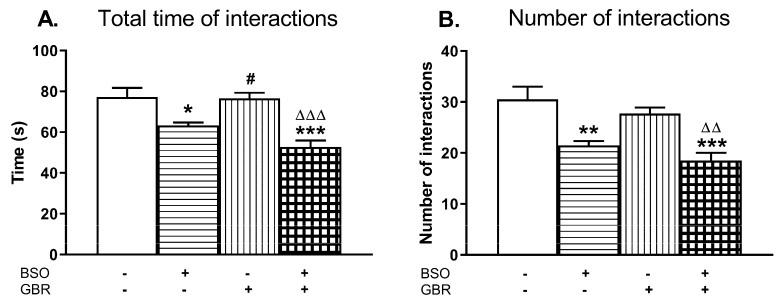
The effects of chronic administration of BSO and GBR 12909, alone or in combination, during the early postnatal life on the social behavior assessed as the total time spent in social interactions (**A**) and as the number of interactions (**B**) in adult, 90-day-old rats. Data are presented as the mean ± SEM, *n* = 12 (six pairs) for each group. Statistical analysis was performed using a two-way ANOVA; symbols indicate significance of differences according to the Newman–Keuls post-hoc test, **p* < 0.05, ***p* < 0.01, ****p* < 0.001 vs. control; ^#^*p* < 0.05 vs. BSO-; and ^∆∆^*p* < 0.01, ^∆∆∆^*p* < 0.001 vs. GBR 12909-treated groups.

**Figure 7 molecules-24-04253-f007:**
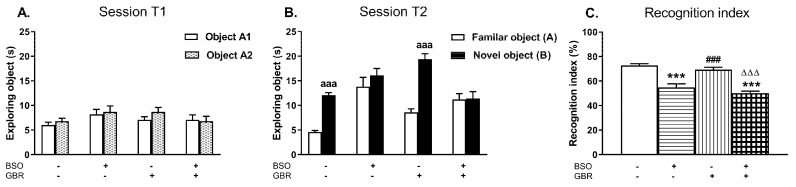
The effects of chronic administration of BSO and GBR 12909, alone or in combination, during the early postnatal life on cognitive functions assessed in adult 91-day-old rats in the novel object recognition test (NOR) test. (**A**) Effects of the examined model compounds on exploration of two identical objects in the acquisition trial (session T1). (**B**) Effects of the examined model compounds on exploration of a novel and a familiar object in the retention trial (session T2). (**C**) Effects of the examined model compounds on the recognition index. Data are presented as the mean ± SEM, *n* = 10 for each group. Letters indicate statistically significant differences between the exploration time of a familiar and a novel object in the session T2 within each studied group, according to the Student’s *t*-test for independent samples, ^aaa^*p* < 0.001 vs. object A. Statistical analysis of recognition index was performed using a two-way ANOVA; symbols indicate significance of differences according to the Newman–Keuls post-hoc test, ****p* < 0.001, vs. control; ^###^*p* < 0.001 vs. BSO-; and ^∆∆∆^*p* < 0.001 vs. GBR 12909-treated groups.

**Figure 8 molecules-24-04253-f008:**
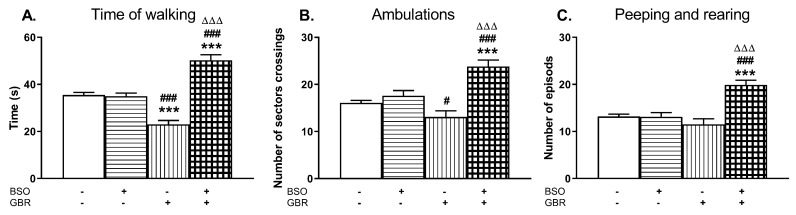
The effects of chronic administration of BSO and GBR 12909, alone or in combination, during the early postnatal life on the exploratory behavior in the open field measured in the adulthood; (**A**) time of walking, (**B**) ambulation, (**C**) peeping, and rearing. Statistical analysis was performed using a two-way ANOVA; symbols indicate significance of differences according to the Newman–Keuls post-hoc test, ****p* < 0.001, vs. control; ^#^*p* < 0.05, ^###^*p* < 0.001 vs. BSO-; and ^∆∆∆^*p* < 0.001 vs. GBR 12909-treated group.

**Figure 9 molecules-24-04253-f009:**
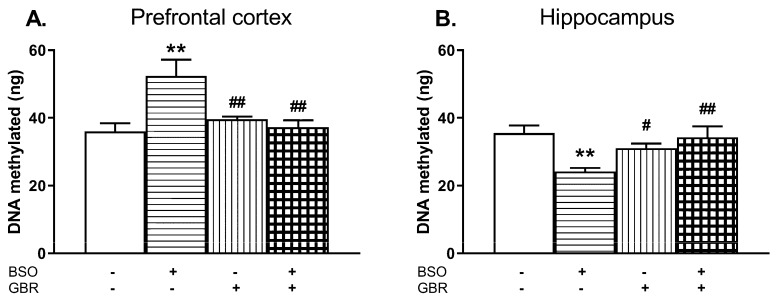
The effects of chronic administration of BSO and GBR 12909, alone or in combination, on the global DNA methylation in the PFC (**A**) and hippocampus (HIP; **B**) of adult 93-day-old rats. Statistical analysis was performed using a two-way ANOVA; symbols indicate significance of differences according to the Newman–Keuls post-hoc test, ***p* < 0.01, vs. control and ^#^*p* < 0.05, ^##^*p* < 0.01 vs. BSO-treated group.

**Figure 10 molecules-24-04253-f010:**
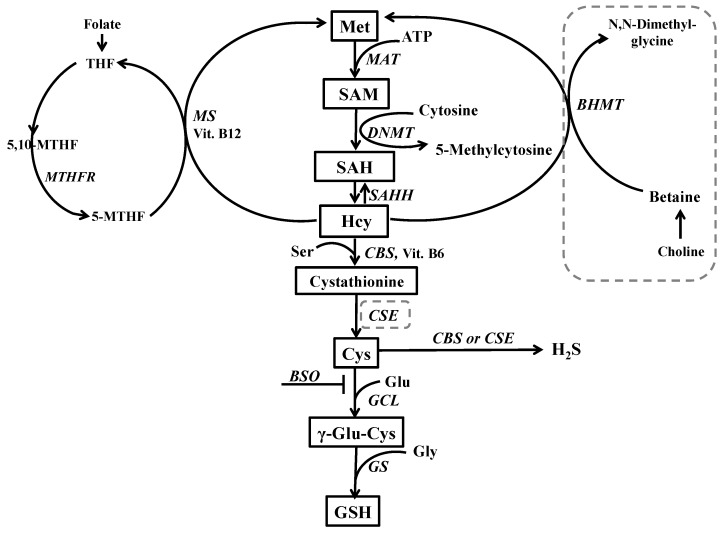
Hepatic metabolic pathways of Met. In the liver, dietary Met is activated by conversion to SAM in an ATP-dependent reaction catalyzed by methionine adenosyltransferase (MAT) SAM (approximately 98%) is preferentially used as a donor of methyl groups for different methyltransferases that transfer them to a variety of acceptor substrates (nucleic acids, proteins, phospholipids, and neurotransmitters) with the simultaneous formation of S-adenosylhomocysteine (SAH) that is further hydrolyzed, by S-adenosylhomocysteine hydrolase (SAHH) to yield Hcy and adenosine. The latter reaction is readily reversible and may cause a chronic elevation of Hcy level. Under normal conditions, Hcy can be either converted into cystathionine by the transsulfuration pathway or used again for the synthesis of Met. Cystathionine is converted to Cys via metabolic reactions mediated by cystathionine β-synthase (CBS) and cystathionine γ-lyase (CSE), consecutively. In the liver, there are two separate remethylation reactions catalyzed by betaine homocysteine methyltransferase (BHMT) and methionine synthase (MS). In the liver, methyl groups used for Hcy re-methylation are provided by both methyl tetrahydrofolate (MTHF) and by the oxidative metabolite of choline, betaine, whereas in other tissues only MTHF is used as a donor of methyl group for MS. The dashed line indicates the lack or weak expression of a given enzyme in the brain. Abbreviations: DNMT–DNA methyltransferase; GCL–γ-glutamate-cysteine ligase; GS—glutathione synthetase; THF—tetrahydrofolate, Ser—serine; Vit. B6—vitamin B6; and Vit. B12—vitamin B12.

**Figure 11 molecules-24-04253-f011:**
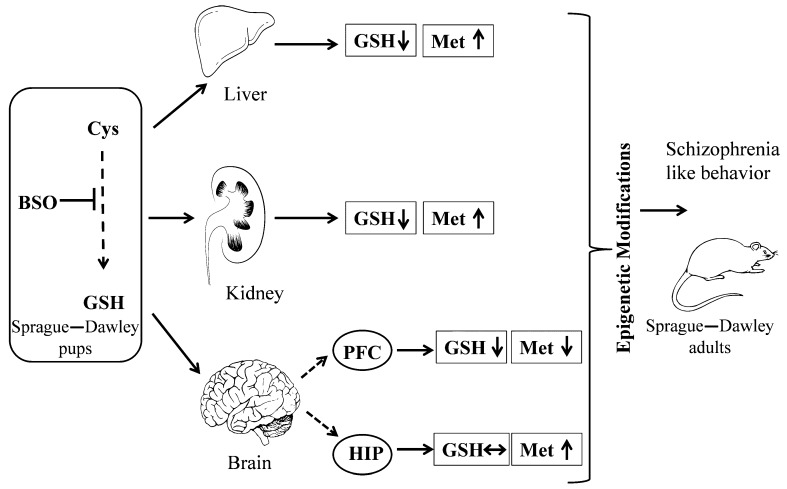
Graphic illustration of the experimental procedure and effects of chronic BSO administration at postnatal days 5–16 on the levels of GSH and Met in 16-day-old rats and on the epigenetic status in the PFC and HIP in adulthood. Arrows indicate: **↑**—increase; **↓**—decrease; and **↔** —no changes.

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
