# Peer review of "Glutathione Deficiency and Alterations in the Sulfur Amino Acid Homeostasis during Early Postnatal Development as Potential Triggering Factors for Schizophrenia-Like Behavior in Adult Rats"

_molecules, 2019, doi:10.3390/molecules24234253_

Round 1

Reviewer 1 Report

The work presented here investigating the pathophysiology of schizophrenia is well presented and the work shows good rigour. 

There are no significant things to address, but there are alot of little things that need to be addressed to increase accuracy 

The abstract could be shortened   The introduction is also very long - 59 references is alot even for a paper this comprehensive.   Minor things to look at - GSH is never defined in the text(Line 6 is first use) line 103 and 104 'via' should be in italics line 103 'Hcy' should be defined line 104 'what is more' is a strange turn of phrase line 192  ANOVA should be defined    The space between 'x %' and 'x%' is different - needs to be consistent  Line 263 should be 'Figure 5A-B', same on line 286 for '6' etc   Method section should be 'mL' not 'ml' and others including line 754 '410 nm''# instead of the correct '410 nM'. There are many examples and this needs to be addressed.   line 714 'bis' should be in italics   I would be interested to see example HPLC traces included in the supporting information if they are available.   The manuscript needs a read through and error checking, but support the publication of this work.

Author Response

Response to Reviewer 1

The whole manuscript has been checked by a professional translator.

The Introduction has been shortened, especially the paragraph on behavioral research. All introduced modifications are marked in yellow.

The abbreviation GSH was explained for the first time in l. 60 of the Introduction.

In lines 91 and 92 and in the whole text the word "via" has been marked with italic (highlighted in yellow).

The extensions for abbreviations Cys (line 90) Hcy (line 91), and Met (92) have been added in the current version of the Introduction.

In line 92 the expression "what is more" has been replaced by the word "moreover".

In line 145 for the first time a two-way ANOVA has been explained as a two-way analysis of variance (ANOVA).

The space between 'x % and 'x% has been made uniform in the whole text (yellow highlighted).

The cited Figures 5A, 5B and figures 6A, 6B have been changed into Figure 5 A-B and Figure 6 A-B.

In the Materials and Methods section, the measure of volume “ml” was replaced by “mL” in the whole text (highlighted in yellow).

The absorbance is always measured at nm but not in nM.

The typical examples of HPLC traces have been included in the Supplementary Materials.

Reviewer 2 Report

The manuscript describes attempts to establish connections between GSH synthesis and dopaminergic transmission, with the two molecular processes standing as important factors in the pathophysiology of schizophrenia. Through the work there is an attempt to assess the effects of L-buthionine-(S,R)-19 sulfoximine (BSO), a GSH synthesis inhibitor, and GBR 12909, a
dopamine reuptake inhibitor, administered alone or in combination, to Sprague-Dawley rats during early postnatal development (p5-p16), on the levels of GSH, sulfur amino acids, global DNA methylation and schizophrenia-like behavior. To that end, the specific organs are examined with respect to the effects of the above two inhibitors (BSO and GBR 12909), finally
projecting those effects to the levels of GSH, Met, Syc, and Hcy. The levels of the aforementioned molecules were assessed for samples from the liver, the kidney, and in the prefrontal cortex (PFC) and hippocampus (HIP) of 16-day-old rats. Concurrently, DNA methylation in the PFC and HIP and schizophrenia-like behavior were assessed in adult rats (p90-p93). The work was done competently with a considerable body of data produced. In the
process, several concerns arose that are addressed below for further consideration by the authors prior to any commitment.

The manuscript needs to undergo a linguistic overhaul. There are several errors in the articulation of the statements related to the results and discussion of the work. The introduction is too long. The authors should briefly introduce the thematic area of work and dwell on the state of the art prior to addressing the issues linked to the research at hand. In the hippocampus case (lines 233-242), the correspondence between what is written and described and what is shown in the graphs of Figure 4 is quite hazy. The section should be written more carefully and clearly. Since the aforementioned sulfur-containing amino acids and derivatives thereof are important in the metabolic physiology of the cells and organisms tested, then concurrent administration of those individually and/or in pairs selected - so as to enhance the effect of biosynthesis or reduction of their levels - might have had a positive effect, thus averting pathological demise. The same holds true for dipeptides (dipeptide derivative of GSH) not containing sulfur or in the presence of bioavailable sulfur. In the discussion section, the authors dwell on the interconversion of the sulfur-containing amino acids cysteine, methionine, homocysteine, with the facile conversion of one toanother, especially the one containing an extra methylene group i.e. HCy. To that end, it
is not clear whether under the presently employed experimental conditions, HCy levels or changes thereof relate to oxidative stress contributing to schizophrenia physiology (from liver to brain). By analogy to statements pertaining to the introduction of the manuscript, the discussion is extremely long. However informative it might be and delineating through extensive citation of reference work, describing the issues at hand, the discussion section should change and become more flexible. To that end, it is strongly suggested that it be compartmentalized into smaller more cohesive sections, autonomously projecting facts and discussion thereof. Thus, small sections could provide an evolving scheme of descriptive results-discussion, ultimately linking through into a final conclusion section. That will help the train of thoughts and the reader to follow through the statements made and the messages intended by the authors. A more concise discussion on the connection of the pro-oxidant/antioxidant status of the PFG and HIP should be provided through the results obtained here so as to project the effect
on schizophrenia. On the basis of the above remarks, the manuscript should undergo extensive revisions and resubmitted for re-evaluation.

Author Response

Response to Reviewer 2

As suggested by the reviewer, the manuscript was checked by a professional translator.

The Introduction chapter has been shortened, especially the paragraph on behavioral research. All introduced modifications are marked in yellow.

In the Results chapter, the description of two-way ANOVA and post-hoc analysis in the hippocampus have been improved. In addition, headers have been added above Figures 1-4 to facilitate tracking of the described results and their further interpretation.

The Discussion chapter has been reorganized and divided into smaller, more coherent sections. Due to the modifications introduced, some parts of the text have been moved to different locations. Every effort has been made to shorten the text of the discussion and make it clearer. Therefore, the first two paragraphs of the introduction have been removed from the original version of the manuscript. The list of references has been shortened from 115 to 107, even though 4 items have been added (items 78-81). All modified or added sentences are marked in yellow. Moreover, in the final part of Discussion, there is a diagram illustrating the main conclusion of this study.

The impact of metabolic disturbances in sulfur compounds on the antioxidant homeostasis in the PFC and HIP has not been discussed in the present study due to the abundance of current data, but a manuscript discussing this issue in different tissues of 16-day-old rats is currently being prepared for publication.

Round 2

Reviewer 2 Report

Following the corrections made by the authors, the manuscirpt could now be considered.

Author Response

According to Reviewer 2 and Editor suggestions, modification to some of the sentence structures used, have been introduced. In the current version of manuscript only the modified sentences have been highlighted yellow.